# ORCA: INTERPRETING PROMPTED LANGUAGE MODELS VIA LOCATING SUPPORTING EVIDENCE IN THE OCEAN OF PRETRAINING DATA

## ABSTRACT

Prompting large pretrained language models leads to strong performance in a variety of downstream tasks. However, it is still unclear *from where* the model learns task-specific knowledge, especially in zero-shot setups. In this work, we propose a novel method ORCA to identify evidence of the model's task-specific competence in prompt-based learning. Through an instance attribution approach to model interpretability, by iteratively using gradient information related to the downstream task, ORCA locates a very small subset of pretraining data that directly supports the model's predictions in a given task; we call this subset *supporting data evidence*. We show that supporting data evidence offers new insights about the prompted language models. For example, in the tasks of sentiment analysis and textual entailment, BERT shows a substantial reliance on BookCorpus—the smaller corpus of BERT's two pretraining corpora—as well as on pretraining examples that mask out synonyms to the task labels used in prompts.[1]

## 1 INTRODUCTION

Large language models (LLMs) are trained on massive text corpora from the web, referred to as the pretraining data (e.g., Devlin et al., 2019; Raffel et al., 2020). Due to their volume, pretraining data typically cannot be inspected manually and are prone to spelling/logic errors, domain mismatch w.r.t. target tasks, social biases, and other unexpected artifacts (Bender et al., 2021). Yet, LLMs pretrained with such noisy data attain surprisingly good performance on numerous downstream tasks, with little or no task-specific tuning (Petroni et al., 2019; Brown et al., 2020).

There are several hypotheses explaining the power of pretrained LLMs. One hypothesis is that the pretraining data is huge and the model *might* be shallowly memorizing patterns in data (Bender et al., 2021; Carlini et al., 2021; Razeghi et al., 2022). An alternative hypothesis is that LLMs *might* be learning to reason through observed patterns in the pretraining data in novel ways (McCoy et al., 2021). However, the *evidence* of these conjectures, especially in arbitrary downstream tasks, remains underexplored.

Such evidence is useful as it can help explain model decisions, surface problematic patterns in data or model behavior, and shed new light on how to improve the model and data (Zhong et al., 2019; Han & Tsvetkov, 2020; 2021; Pruthi et al., 2022). Moreover, it can facilitate the trustworthiness of the models (Lipton, 2018; Jacovi et al., 2021).

In this work, we develop a methodology to provide such evidence. Our hypothesis is that among the enormous pretraining corpora, there is a subset of pretraining data that contributes to the model's behavior on a downstream task more than the rest of the pretraining data. Therefore, our task is to locate a task-specific evidence set—a very small amount of pretraining data that particularly helps the model's performance on the task. We call it *supporting data evidence* (SDE). Such SDE can help interpret the model if we analyze its task-relevant patterns compared to the rest of the corpora.

A related line of interpretability research focuses on instance attribution (Koh & Liang, 2017; Yeh et al., 2018; Pruthi et al., 2020; Han et al., 2020), where the goal is to find which training examples

---

[1]Code and data will be released at `ANONYMIZED` upon publication.

are most influential to the model's decision, focusing on individual test examples. However, in this work we are interested in locating sets of pretraining data influencing the *whole task* (i.e., a full test set, rather than individual test instances). We seek such "global" evidence for the task because given the scale of the pretraining and task data, it could be inefficient or even infeasible to find and inspect the evidence for each of the task examples.[2]

We first formulate the problem of finding SDE in pretraining data by upweighting the SDE set and measuring its impact on model performance (§2.1). In §2.2, we propose a novel method ORCA[3] that effectively identifies the SDE by iteratively using task-specific gradient information. On two classification tasks—sentiment analysis and textual entailment—in a prompt-based setup (§3), we show the effectiveness of the SDE discovered by ORCA compared to random data subsets and nearest-neighbor data in an embedding space (§4). Our analyses of the discovered SDE (§5) show that our base language model BERT (Devlin et al., 2019) has an interestingly high reliance on the smaller corpus of its two pretraining corpora (BookCorpus, Zhu et al., 2015). Also the pretraining examples in SDE typically mask out synonyms to the task verbalizers (i.e., words mapped to the task labels in the prompts, Schick & Schütze, 2021).

## 2 ORCA 🐋

We develop a method to explain the competence of large pretrained language models used in zero- or few-shot prompt-based classification (Petroni et al., 2019; Brown et al., 2020).[4] Without conventional finetuning, model decisions rely on knowledge learned from the pretraining data, and our goal is to identify what supporting data evidence (SDE) in pretraining data facilitates model's competence in a specific downstream task.

### 2.1 PROBLEM FORMULATION

Assume $\theta^{\text{PT}}$, a LLM pretrained with a dataset $D^{\text{PT}} \ni (x_{\text{context}}^{\text{PT}}, y_{\text{masked}}^{\text{PT}})$. For example, for a masked language model $x_{\text{context}}^{\text{PT}}$ is a block of text with certain tokens masked, and $y_{\text{masked}}^{\text{PT}}$ are the masked tokens in their original forms, to be reconstructed. $\theta^{\text{PT}}$ is trained to minimize a loss $\mathcal{L}$ over the pretraining examples, $\theta^{\text{PT}} = \arg\min_\theta \mathcal{L}(D^{\text{PT}}; \theta)$.

The LLM can be applied to many downstream tasks without finetuning, via prompting (Schick & Schütze, 2021; Liu et al., 2021). Given a dataset in a downstream classification task $D^{\text{task}} \ni (x^{\text{task}}, y^{\text{task}})$, the LLM is applied by measuring $p_\theta(\text{verbalizer}(y^{\text{task}}) \mid \text{template}(x^{\text{task}}))$. The template supplies a prompt tailored to the task for the model, and the verbalizer maps the output of the language model to the task's label space (more details in §3.2).

We interpret the model decisions by finding the SDE $S \subset D^{\text{PT}}$ w.r.t. the task data $D^{\text{task}}$. The size of $S$ should be very small (e.g., a few hundred) compared to the whole pretraining data, $|S| \ll |D^{\text{PT}}|$, to facilitate further manual or semi-automatic analyses. More importantly, $S$ should "contribute" significantly to the performance of the model on the downstream task.

However, we first observe that defining this contribution is a non-trivial problem. Prior work in instance attribution like influence functions (Koh & Liang, 2017) adopts a "leave-one-out" perspective (Cook, 1977). In our case, this would mean removing $S$ from $D^{\text{PT}}$, retraining a new LLM from scratch, and testing it on $D^{\text{task}}$. This is prohibitively expensive.[5]

We adopt an "upweighting" perspective. Instead of leave-one-out, we upweight certain pretraining examples (e.g., $S$) by training the model on these examples for an additional epoch. The resulting change to the model should be small to prevent overfitting. Specifically, we randomly batch the SDE $S$ to mini-batches, thereby updating the model via a very small number of optimizer updates:

$$\theta_{\text{new}}^{\text{PT}} \leftarrow \theta^{\text{PT}} + updates_{\theta, \mathcal{L}}(S) \tag{1}$$

---

[2]Directly applying instance attribution methods to the task level has also been shown to yield negative results (Kocijan & Bowman, 2020).

[3]Named after the marine mammal for n**O** pa**R**ti**C**ular re**A**son.

[4]While in this work we focus on text classification, the framework is also adaptable to generation problems.

[5]Moreover, the definition of influence functions and even leave-one-out can sometimes be arguable, especially in non-convex models (Basu et al., 2021; K & Søgaard, 2021).

For the simplicity of notation, we fold a sequence of optimizer updates into $updates_{\theta,\mathcal{L}}$ that depends on a sequence of batched data $S$, the model parameters $\theta$, the loss function $\mathcal{L}$, and an optimization algorithm used during pretraining.

The quality of the data evidence is reflected in the *performance difference* in the downstream task between the new model $\theta_{new}^{PT}$ and the original pretrained model $\theta^{PT}$, measured by the metric associated with the task.[6]

## 2.2 IDENTIFYING SUPPORTING DATA EVIDENCE

We now propose a novel method ORCA identifying the supporting data evidence $S \subset D^{PT}$. The goal is to find a subset of the pretraining data $|S| \ll |D^{PT}|$ that is directly helpful to the downstream task when we continue pretraining the language model over it (as described in §2.1). The intuitions behind our method are simple: (1) We aim to find contributive examples in $D^{PT}$ that exert a similar change to the model parameters as $D^{task}$ would. (2) There could be multiple subsets of pretraining data that, in conjunction, are useful to the task. We thus select $S$ in several iterations rather than at once.

We build our SDE $S$ in $m$ iterations, $S_1, S_2, \ldots, S_m$; the size of the subset at each iteration is $\frac{|S|}{m}$. To find the first evidence subset $S_1$, we rely on the intuition that continuing training on the task data $D^{task}$ directly is likely to improve the original model $\theta^{PT}$ on the task. We thus batch the task data and calculate a batch gradient $\nabla_\theta \mathcal{L}_{task}(D^{task}; \theta^{PT})$.[7] Descending along the gradient direction should improve $\theta^{PT}$, and we find a subset $S_1$ of the pretraining data that exerts a similar gradient of the model as $\nabla_\theta \mathcal{L}_{task}(D^{task}; \theta^{PT})$:

$$S_1 = \{d \in D^{PT} \mid \cos(\nabla_\theta \mathcal{L}_{LM}(d, \theta^{PT}), \nabla_\theta \mathcal{L}_{task}(D^{task}; \theta^{PT})) > \delta_1\} \tag{2}$$

We measure a cosine similarity between the gradient for each example in $D^{PT}$ and the batch gradient for the task.[8] We then select for $S_1$ the top-$k$ examples in $D^{PT}$ with highest cosine; $\delta_1$ is simply the cosine score of the $k$-th ranked example. §3.4 specifies the size of selection along with other hyperparameters.

Now with the first data evidence subset $S_1$, we continue pretraining an intermediate model $\theta_1^{PT}$:

$$\theta_1^{PT} \leftarrow \theta^{PT} + updates_{\theta,\mathcal{L}}(S_1) \tag{3}$$

The procedure to find the rest of the data evidence subset $S_i$ with $i = 2, 3, \ldots, m$ is similar to the above but with one difference: these subsets should ideally be beneficial to the model in a way that is not already fully captured by $S_1$.

We hypothesize that the intermediate model $\theta_1^{PT}$ captures information about $S_1$. Therefore, for later iterations we want to calculate a task batch gradient based on the previous intermediate model, $\nabla_\theta \mathcal{L}_{task}(D^{task}; \theta_{i-1}^{PT})$. The data evidence subset at each iteration should again exert a similar gradient:

$$S_i = \{d \in D^{PT} \mid \cos(\nabla_\theta \mathcal{L}_{LM}(d, \theta_{\lfloor i-1 \rfloor}^{PT}), \nabla_\theta \mathcal{L}_{task}(D^{task}; \theta_{i-1}^{PT})) > \delta_i\} \tag{4}$$

with $\delta_i$ as a threshold for selecting $|S_i|$ elements like $\delta_1$. $\lfloor i-1 \rfloor$ is a design choice that can allow for a "lagged" model (i.e., computing the gradient of the LM loss w.r.t. the model several iterations before vs. the immediate previous intermediate model). This lagging aims to improve the stability of the method. For the experiments in this work, $\theta_{\lfloor i-1 \rfloor}^{PT}$ is by default $\theta^{PT}$, for a maximum lagging; $\theta_{\lfloor i-1 \rfloor}^{PT}$ is $\theta_{i-1}^{PT}$ in cases denoted by *NL* (no lagging).[9]

At each iteration, having a total of $i$ data evidence subsets, we continue pretraining an intermediate model $\theta_i^{PT}$:

$$\theta_i^{PT} \leftarrow \theta^{PT} + updates_{\theta,\mathcal{L}}(\cup_{j=1}^i S_j) \tag{5}$$

---

[6] A discussion over the limitations of our problem formulation can be found in §A.

[7] The task loss over a single task example is $\mathcal{L}_{task}(x^{task}, y^{task}) = -\log p_\theta(\text{verbalizer}(y^{task}) \mid \text{template}(x^{task}))$.

[8] The LM loss over a single pretraining example is $\mathcal{L}_{LM}(x_{context}^{PT}, y_{masked}^{PT}) = -\log p_\theta(y_{masked}^{PT} \mid x_{context}^{PT})$.

[9] Compared to $\theta^{PT}$, the no-lagging version uses $\theta_{i-1}^{PT}$ which comes from continuing pretraining over a small amount of examples and can have a higher variance. Adding this lagging also remotely shares intuition with some RL methods addressing training stability (Mnih et al., 2016).

It is worth noting that for every iteration, we continue pretraining over the *original* language model, and the data evidence subsets are unordered.

After the $m$-th iteration, we complete building our full supporting data evidence $S = \cup_{j=1}^m S_j$. The resulting model $\theta_m^{\text{PT}}$ is essentially the final upweighted model, i.e., $\theta_{\text{new}}^{\text{PT}}$ introduced in §2.1.

## 3 EXPERIMENTAL SETUP

### 3.1 BASELINE METHODS

ORCA is a greedy algorithm and thus not guaranteed to find the global optimal SDE out of the $\binom{|D^{\text{PT}}|}{|S|}$ candidates. To evaluate the efficacy of the identified evidence, we compare ORCA with the following baseline methods:

**Random sampling**   We simply sample at random $|S|$ examples from $D^{\text{PT}}$ as the SDE.

**Embedding nearest neighbors**   Enhancing language models using examples with nearest neighboring embeddings is a common approach in domain adaptation of LMs and kNN-LMs (Gururangan et al., 2020; Khandelwal et al., 2020). Here we find nearest neighboring pretraining examples to the task examples. We define a similarity score as below:

$$\cos(h_{\text{masked}}(\hat{x}_{\text{context}}^{\text{PT}}), h_{\text{verbalizer}}(\text{template}(\hat{x}^{\text{task}}))) \tag{6}$$

- $h_{\text{masked}}$ is the last hidden representation at the position of the masked pretraining token.
- $h_{\text{verbalizer}}$ is the last hidden representation at the position of the task verbalizer token.
- $\hat{x}_{\text{context}}^{\text{PT}}$ is the pretraining input to the model but containing the ground truth masked token.
- $\text{template}(\hat{x}^{\text{task}})$ is the templated task input but supplying the ground truth verbalized label.

We use the ground truth information here for a fair comparison with our method ORCA, where the calculation of gradients involves the ground truth information as well.

Practically, since $|D^{\text{task}}|$ can be large and well over $|S|$, we first sample $t$ examples from $D^{\text{task}}$. Then, for each of the $t$ sampled task examples, we find the top-$k$ nearest neighboring pretraining examples in $D^{\text{PT}}$. Finally, from the pool of the $t \cdot k$ pretraining examples, we sample $|S|$ of them as the SDE. We additionally have a hyperparameter max-$r$ controlling the maximum allowed data repetitions in the selected data evidence.

**Iterative selection with embeddings**   We use the gradient information $\cos(\nabla_\theta \mathcal{L}_{\text{LM}}(.), \nabla_\theta \mathcal{L}_{\text{task}}(.))$ when collecting the SDE subsets in ORCA iteratively. Here we test whether we can substitute the gradients with embeddings while keeping the selection iterative. Reusing the notations in the embedding nearest neighbors baseline, we use $\cos(h_{\text{masked}}(.), \frac{1}{|D^{\text{task}}|} \sum_{x^{\text{task}}} h_{\text{verbalizer}}(.))$ for all the gradient cosine operations in ORCA. Note that we use the average embeddings of all task examples to replace the batch gradient over all task examples. Other design decisions of ORCA remain unchanged.

### 3.2 LANGUAGE MODEL AND DOWNSTREAM TASKS

**BERT**   We use the BERT-large language model as $\theta^{\text{PT}}$ (Devlin et al., 2019).[10]

**IMDB**   We primarily experiment with two text classification tasks, sentiment analysis and textual entailment.[11] For sentiment analysis, we use the IMDB movie review dataset (Maas et al., 2011). The task data $D^{\text{task}}$ here is the IMDB test split containing 25,000 examples.[12] The template for the IMDB examples is "*It was [MASK]. <REVIEW>*". The verbalizer maps the reconstruction of the [MASK] token to the label space, {"good" $\rightarrow positive$, "bad" $\rightarrow negative$}.

---

[10]We choose it primarily due to the limited computing resources we have—BERT is small both in terms of the number of model parameters and the size of the original pretraining data. ORCA is extendable to other language models as well.

[11]Future work can explore more tasks, but we select two typical ones here mainly due to our computational resources (more details in §B).

[12]The use of test set is our deliberate choice in this work. In §D, we further discuss the purpose of it and show a sanity check on an alternative setup using the training set.

**MNLI** For the textual entailment task, we use the MNLI dataset (Williams et al., 2018). The task data $D^{\text{task}}$ here is the MNLI matched validation split containing 9,815 examples (the test split is private). The template for the MNLI examples is "*<PREMISE>* `[MASK]`, *<HYPOTHE-SIS>*". The verbalizer maps the reconstruction of the `[MASK]` token to the label space, {"yes" $\rightarrow$ $entailment$, "no" $\rightarrow$ $contradiction$, "maybe" $\rightarrow$ $neutral$}. We use the OpenPrompt library (Ding et al., 2022) to prompt the BERT model with the templates and verbalizers inherited from Gao et al. (2021b).

**Zero-shot transfer and prompt tuning** When we formulate our problem, we are interested in the evidence in *pretraining* that directly impact the pretrained model's performance on the downstream task—a zero-shot transfer scenario. There is no notion of *finetuning* with the in-task training data. However, research in prompt tuning (e.g., Lester et al., 2021) folds the usage of in-task training data into the template for the task. A sequence of soft embeddings is added to the beginning of the template and trained with the in-task training data. The language model parameters remain unchanged. Apart from our main experiments with the zero-shot transfer model, we also consider such prompt tuning scenarios, finding pretraining data evidence useful for the task when the template is enhanced with some in-task training data.[13]

### 3.3 PRETRAINING DATA

**Source** BERT uses the English *Wikipedia* and *BookCorpus* (Zhu et al., 2015) as its pretraining data. During pretraining, 15% of the tokens are randomly masked out to be reconstructed. Though BERT's pretraining data is already small compared to those of many other language models (e.g., Raffel et al., 2020; Gao et al., 2021a), we unfortunately still do not have the resource to process the full dataset. In fact, in this work we only randomly sample 0.5% of the full pretraining data.

**Format** During pretraining, BERT would reconstruct the masked 15% tokens in a sequence in parallel (i.e., the reconstruction loss for each token is independent). From the training perspective, this is efficient. However, this work aims to find the SDE. We particularly want to know learning the reconstruction of *which* token could most impact the downstream task performance. Therefore, we expand each pretraining data and treat each masked token as a standalone example.[14] More specifically in our setup, $D^{\text{PT}} \ni (x^{\text{PT}}_{\text{context}}, y^{\text{PT}}_{\text{masked}})$. $x^{\text{PT}}_{\text{context}}$ is a sequence of 512 tokens, and $y^{\text{PT}}_{\text{masked}}$ is a single masked token in the sequence. Together this makes $|D^{\text{PT}}| = 3{,}924{,}635$ (with 52,640 unique $x^{\text{PT}}_{\text{context}}$ sequences). We choose at most 2,000 instances from $D^{\text{PT}}$ as the SDE $S$.

### 3.4 HYPERPARAMETERS

ORCA finds $S$ in iterations. In this work we use $m$=20 iterations, with each iteration finding 100 examples from $D^{\text{PT}}$ (with a total $|S|$=2000).

For the embedding kNN baseline, we sample $t$=1000 task examples and choose $k$={10, 20, 50, 100} most similar pretraining data. Within the $t \cdot k$ candidate pool, we sample $|S|$=2000 examples, with a max number of repetitions $r$={1, 20, 2000}.[15]

During the continued pretraining for all methods, we use a batch size of 16, resulting in *at most 125 optimizer updates* from the original language model. The learning rate is set at one of BERT's default values 2e-5.

### 4 EVALUATION

---

[13]We use different amounts of in-task training data for prompt tuning depending on the task performance. For IMDB, we use 100 examples per class, whereas for MNLI, we use 10,000 examples per class.

[14]Other masked tokens are still masked in the input context, to be faithful to the original LM objective.

[15]In our method ORCA, though the examples *within* $S_i$ are strictly non-overlapping, we don't enforce distinctiveness *across* $S_i$ since we only work with 0.5% of the pretraining data. This means an example could at *maximum* appear 20 times in our method. Therefore, we include the $r > 1$ options for the embedding kNN baseline as well for a fairer comparison. In §C we further discuss the choice of $t$ and $k$.

[16]Additional details and discussion of the results can be found in §E.

Table 1: Main results (accuracy) of ORCA and baselines on the zero-shot model. NL means the no-lagging variant. Numbers in regular fonts are averaged from 5 random seeds, while numbers in small fonts show a trajectory of performance with one seed.[16]

| On zero-shot model | IMDB | MNLI |
|---|---|---|
| *Null* | *73.50* | *43.70* |
| **Random** | 71.25 $_{\pm2.56}$ | 42.56 $_{\pm0.43}$ |
| **Embedding kNN** | 76.55 $_{\pm2.16}$ | 45.15 $_{\pm0.48}$ |
| **Iterative embeddings** | 75.11 $_{\pm4.59}$ | 43.74 $_{\pm0.68}$ |
| **ORCA** (NL) | **84.51** $_{\pm0.77}$ | 45.46 $_{\pm0.73}$ |
| $0 < |S| \leq 500$ | 79.81 | 44.85 |
| $500 < |S| \leq 1000$ | 83.87 | 45.64 |
| $1000 < |S| \leq 1500$ | 84.40 | 46.10 |
| $1500 < |S| \leq 2000$ | 85.17 | 46.49 |
| **ORCA** | 84.33 $_{\pm1.51}$ | **46.06** $_{\pm0.35}$ |
| $0 < |S| \leq 500$ | 81.60 | 45.99 |
| $500 < |S| \leq 1000$ | 83.23 | 45.75 |
| $1000 < |S| \leq 1500$ | 84.42 | 46.40 |
| $1500 < |S| \leq 2000$ | 85.15 | 46.26 |

We evaluate the supporting data evidence $S$, identified using ORCA and the baselines, by quantifying the supportiveness of $S$ (as defined in §2.1). Note that this section does not focus on whether or not the discovered data evidence is plausible to humans. We will explore what humans can interpret from the actual data evidence in §5.

Table 1 shows our main results: the performance of our zero-shot language model pretrained additionally on $S$, as identified by different methods. We first notice a performance gap between our original model on IMDB and MNLI, indicating the entailment task is intrinsically harder for models that have only been trained on pretraining data. We observe a moderate performance improvement using the embedding nearest neighbors method ($|S| = 2000$). The best performance is achieved by our proposed method ORCA, especially in the task of IMDB by a large margin (even with $|S| \leq 500$), showing the effectiveness of our method.

Table 2 shows some additional results on the effect of the SDE $S$ on a prompt-tuned model. These results show that, compared to the zero-shot model, a prompt-tuned model is more difficult to improve since the prompt may already be highly specialized towards the task, using the in-task training data. The *additional* signals in the pretraining data that are useful to the task can be scarce. That said, the pretraining data $S$ identified by ORCA still improves the model on IMDB.

## 5 ANALYSIS

While useful in showing the effectiveness of the SDE $S$, evaluations in §4 do not provide us with tangible insights about the model itself. In this section, we analyze some properties of $S$, and see whether they reflect humans' expectations for the model. We first show a few qualitative examples of the evidence discovered by ORCA in Table 3.

**Which source corpus does the supporting data evidence come from?** The pretraining data of BERT consist of the English Wikipedia and BookCorpus. We show the source corpus of examples in $S$ in Figure 1.

Table 2: Additional results (accuracy) of ORCA and the baselines on the prompt-tuned model. All the numbers are averaged from 5 random seeds.

| On prompt-tuned model | IMDB | MNLI |
|---|---|---|
| *Null* | *87.83* | *70.19* |
| **Random** | 86.06 $_{\pm0.82}$ | 69.07 $_{\pm0.50}$ |
| **Embedding kNN** | 86.53 $_{\pm0.77}$ | 68.93 $_{\pm0.61}$ |
| **Iterative embeddings** | 87.80 $_{\pm0.03}$ | 68.45 $_{\pm0.36}$ |
| **ORCA** (NL) | 87.65 $_{\pm1.10}$ | 68.79 $_{\pm0.28}$ |
| **ORCA** | **88.10** $_{\pm0.65}$ | 68.61 $_{\pm0.44}$ |

We find that though the pretraining set consists of considerably more data from Wikipedia than from BookCorpus (76.5% vs. 23.5%), the SDE identified by ORCA has a drastically different source corpus distribution. In IMDB, 64.1% and 92.6% of the examples in $S$ come from BookCorpus, using the default ORCA and its no-lagging variant respectively. The demotion of Wikipedia examples in the sentiment analysis task is somewhat reasonable, since Wikipedia is meant to have a neutral point of view (NPOV).[17] On the other hand, BookCorpus consists of novels that could involve strong emotions and sentiments.

---

[17]https://en.wikipedia.org/wiki/Wikipedia:Neutral_point_of_view

Table 3: Examples of the supporting data evidence ($S$) in pretraining data discovered by ORCA for IMDB and MNLI. The masked token ($y_{\text{masked}}^{\text{PT}}$) in each example is underlined. The example evidence for IMDB expresses sentiments, while it is less clear whether the example evidence for MNLI is related to entailment.

| | |
|---|---|
| IMDB | ... we have to think that were awfully lucky as human beings to have the nice precise system. the sloppy system is probably **good** enough for bacteria. it turns out – much to geneticists surprise – that lowly bacteria store genes as whole units (weasel) ... |
| | it was the only place she could afford. her meager earnings didnt provide much in the ways of clean, modern style along with the privacy she required. she felt better if she thought about how **bad** it could be. a year ago, shed lived with her mother. anywhere was better than living with her ... |
| MNLI | ... he then cut the cord that bound her hands and legs. are you ok to walk? he asked hoping the answer was **yes**. i think so. but im quite stiff, she said. he helped her up. stretch your legs a little. theyll feel better ... |
| | ... there was no way to hide the shock on her face, and she knew he saw it by his sigh. "do you think yourself less than me?" "**no**!" she absolutely didn't but ... he nodded his head. "i see. you thought i would think you were less than me." she was ashamed. "i'm sorry." ... |
| | ... he shook his head, incredulous. in fact, he looked like he was considering throttling me. "you're just not getting it. maybe that's my fault. **maybe** it's because i don't tell you i love you often enough. baby, you're the only 'good' thing that i've ever had ... |

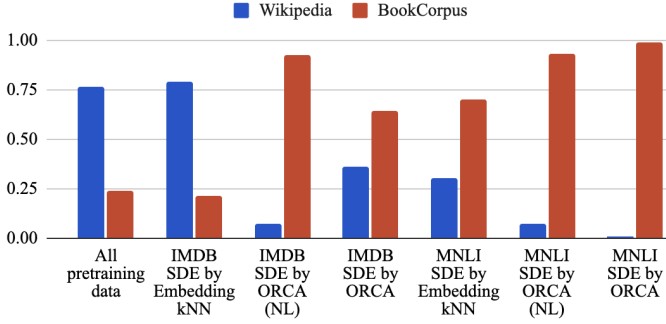

Figure 1: Source corpus distribution of the supporting data evidence (SDE) in IMDB and MNLI.

A similar trend can be found in MNLI as well, with 99.0% and 92.9% of the examples in $S$ coming from BookCorpus, using the default and NL variant of ORCA. We conjecture that the over-reliance on BookCorpus in MNLI could be due to the selection of the colloquial verbalizer words (e.g., "yes", "maybe"), which can be scarce in Wikipedia. Also, the BookCorpus data could contain more everyday topics that match MNLI's genres (e.g., fiction, letters, telephone speech). However, whether it is reasonable for the model to rely on BookCorpus for textual entailment is arguable: Wikipedia should be a more reliable source if we want the model to build more upon factual information.

**What are the masked tokens in the supporting data evidence?** Prompted language models use a verbalizer to adapt to the downstream task. For example, outputting "good" for a templated IMDB input indicates a positive sentiment, "yes" for MNLI indicates entailment, etc. For a pretraining example that supports the task, are there any relations between its masked, to-be-reconstructed pretraining token ($y_{\text{masked}}^{\text{PT}}$) and the verbalizer words for the task (verbalizer($y^{\text{task}}$))? In Table 4, we show the 10 most frequent masked words (types) in $S$, for each method in IMDB and MNLI.

We observe that the verbalizer words, in their original forms, are always the most common masked token in $S$. For all of the methods in both tasks, over 50% of the masked tokens are exactly the verbalizer words. Though there is some noise in $y_{\text{masked}}^{\text{PT}}$ (e.g., symbols that carry no task-relevant meaning), most of the other masked tokens are synonyms to the verbalizer words in IMDB by observation. In MNLI, the other masked tokens may capture relations between clauses similar to the verbalizer words (e.g., then, to, probably). Overall, we find that $y_{\text{masked}}^{\text{PT}}$ in the discovered $S$ is reasonable for the sentiment analysis and textual entailment task.

Table 4: Masked tokens ($y^{\text{PT}}_{\text{masked}}$) in the supporting data evidence of IMDB and MNLI.

| Task | Method | Most frequent $y^{\text{PT}}_{\text{masked}}$ in $S$ |
|------|--------|-----------------------------------------------------|
| IMDB | Embedding kNN | **bad**, **good**, terrible, great, badly, excellent, worst, negative, better, disappointment, ... [*11 distinct tokens in total, 94.8% **verbalizer words***] |
|      | ORCA (NL) | **bad**, **good**, worst, n, worse, ', wrong, -, horrible, poisonous, ... [*91 distinct tokens in total, 90.0% **verbalizer words***] |
|      | ORCA | **bad**, **good**, `, horrible, not, worse, ugly, hated, poor, terrible, ... [*285 distinct tokens in total, 55.9% **verbalizer words***] |
| MNLI | Embedding kNN | **no**, **yes**, **maybe**, `, yeah, However, perhaps, n, ), No, ... [*258 distinct tokens in total, 58.3% **verbalizer words***] |
|      | ORCA (NL) | **maybe**, **yes**, **no**, `, n, that, -, then, perhaps, the, ... [*176 distinct tokens in total, 69.1% **verbalizer words***] |
|      | ORCA | **maybe**, **yes**, **no**, `, perhaps, to, probably, has, in, big, ... [*125 distinct tokens in total, 59.4% **verbalizer words***] |

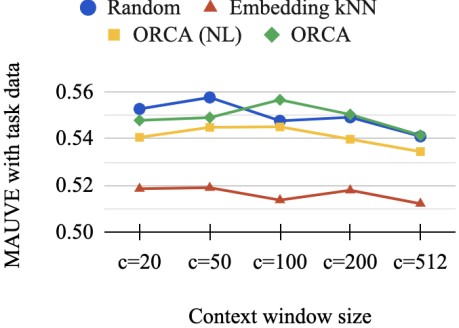

Figure 2: MAUVE similarity on **IMDB**, between the sets of $x^{\text{PT}}_{\text{context}}$ in $S$ and $x^{\text{task}}$.

**Is the context of the supporting data evidence similar to the task input data?** We are interested in the relationship between the context of the selected pretraining data ($x^{\text{PT}}_{\text{context}}$) and the input of the downstream task ($x^{\text{task}}$). Are they exceptionally similar, indicating that the model may be memorizing shallow patterns? Alternatively, are they considerably different, indicating that the model needs to transfer some learnt knowledge from pretraining to the task (either in a reasonable or spurious way)? Our exploratory step uses an automatic metric between two distributions of texts, MAUVE (Pillutla et al., 2021), to measure the similarity between our sets of $x^{\text{PT}}_{\text{context}}$ and $x^{\text{task}}$. As a method based on quantized language model embeddings, MAUVE similarity may capture text attributes such as topics and style.[18]

Apart from using all 512 tokens in the context of the data evidence ($x^{\text{PT}}_{\text{context}}$), we also truncate the context, keeping the surrounding $c$ tokens of the masked token ($y^{\text{PT}}_{\text{masked}}$). We control for the scope of the context by varying $c$. For $x^{\text{task}}$, we randomly sample 2000 examples to match the size of $S$.[19] Figure 2 shows the results on IMDB. See §F for MNLI results.

We observe that the MAUVE scores between $x^{\text{PT}}_{\text{context}}$ and $x^{\text{task}}$ are all between 0.512 and 0.577. In contrast, the MAUVE score between the training set of the task and the test set ($x^{\text{task}}$) is 0.998 and 0.997 for IMDB and MNLI respectively. This substantial difference in MAUVE scores may indicate a disparity in topics and style between the context of the pretraining evidence and the task data. Additionally, the MAUVE score of our selected SDE is not higher than a random sample in most cases. This further shows that the signal in the evidence context useful for the task is *subtle*, in a way that MAUVE cannot capture. This is in contrast to the conjecture that the model must have seen the exact inputs in the pretraining data and is only performing a shallow memorization.

While not within the scope of this paper, future investigation can also extend the analysis of the supporting evidence with feature attribution methods (Pezeshkpour et al., 2022) or a human evaluation with domain experts of the task (e.g., what exact spans in the SDE contribute to their supportiveness). We further discuss the limitations of our method, computational resources, and future directions in §A and §B.

---

[18]Grammaticality can be another attribute as Pillutla et al. (2021) work with machine-generated texts. This is less relevant in our case as our sets of texts are naturally occurring.

[19]Here $x^{\text{task}}$ is without template, and $x^{\text{PT}}_{\text{context}}$ has the masked token recovered. This is to give MAUVE most natural texts for evaluation.

## 6 RELATED WORK

LLMs have been showing competence in various downstream tasks in NLP with little to no task-specific tuning, using prompts (Petroni et al., 2019; Brown et al., 2020; Schick & Schütze, 2021; Gao et al., 2021b; Lester et al., 2021). We are especially interested in interpreting LLMs under a zero-shot setup, where the knowledge relevant to the downstream task must come from the noisy pretraining data.[20]

One common interpretability method for NLP models is feature attribution, where important tokens or spans in the inference-time input are highlighted, indicating their contributions to the model's decision (Simonyan et al., 2014; Li et al., 2016; Ribeiro et al., 2016; Lundberg & Lee, 2017). However, information relevant to explaining the model's decisions (especially if abstract) is often not in the inference-time input (Han et al., 2020; Wiegreffe & Marasović, 2021; Pezeshkpour et al., 2022). On language models, feature attribution has been used to interpret and verify grammatical phenomena (Yin & Neubig, 2022).

Another type of interpretation that aligns more with our focus is instance attribution, where important training examples are highlighted for their influence on the model (Koh & Liang, 2017; Yeh et al., 2018; Pruthi et al., 2020; Han et al., 2020; Guo et al., 2021). In this work, we are instead interested in the influence of pretraining data and in finding SDE for the entire task rather than individual test examples.[21] There has also been prior work analyzing what amount of data is needed during pretraining to achieve models with certain capabilities (Zhang et al., 2021), but these works do not attribute model performance to specific pretraining data.

Our formulation may seem similar to prior work in task-enhancing pretraining (Han & Eisenstein, 2019; Gururangan et al., 2020; Yao et al., 2021). However, such methods typically use a large amount of loosely relevant pretraining data along with the in-task training data, to improve performance. We instead aim to find an orders-of-magnitude-smaller set of pretraining data, providing a clearer signal of their impact on the task for interpretability purposes.

Our proposed method to find the data evidence, ORCA, shares a similar intuition with prior work that reweighs training data (Wang et al., 2020), as both methods use the gradient information of the test data. However, their target model depends on an *ordered sequence* of data weights and model checkpoints. In contrast, we apply an *unordered* data evidence set to the *original* model, mimicking an upweighting in pretraining. In addition to the difference in methods, the purpose is different as well: theirs is performance-oriented while ours is interpretability-oriented.

Other remotely related line of work in machine learning includes coreset construction (Coleman et al., 2020; Mirzasoleiman et al., 2020; Huang et al., 2021) and dataset distillation (Wang et al., 2018; Zhao et al., 2021). Their focus is typically an empirical risk minimization problem on the training data, without a notion of downstream tasks or task transfer. They aim to create a substitution set for the full training data for an efficiency purpose.

## 7 CONCLUSION

The source of competence of zero- and few-shot prompted language models on downstream tasks is mysterious. The models should be gaining task-specific knowledge from the pretraining data, but what pretraining data leads to the capability of the models is an underexplored area of research. In this work, we formulate the problem of finding supporting data evidence in the pretraining data of LLMs for downstream tasks. We propose ORCA to effectively identify such evidence with an iterative guide from task-specific gradient information. Deeper analyses into the evidence show that a prompted BERT on sentiment analysis and textual entailment relies heavily on the BookCorpus data, as well as on pretraining examples that mask out task verbalizers and their synonyms.

---

[20]Interpreting the role of pretraining data in an unprompted, finetuning setup is intrinsically harder, but prior work like Chen et al. (2020) have made attempts.

[21]A concurrent work by Akyürek et al. (2022) builds a candidate set for fact-tracing in question answering; the difference is the use of task-related training examples instead of pretraining data, and an information retrieval evaluation.

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

## A    LIMITATIONS OF THE PROBLEM FORMULATION

**Implication of supporting data evidence**   One limitation of our setup is that the data outside our defined SDE $S$ may have additional value on the model which the current formulation does not capture. For instance, there could be certain examples that exceptionally help the model capture the grammar of the language, making an indirect contribution to the task. However, they are rather unlikely to be picked as SDE: in the scope of this paper, we focus on finding the evidence that is *directly* related to the downstream task. The indirect evidence is also interesting and may be valuable for future work to investigate further (e.g., via finding which examples contribute most to the *contribution of SDE*).

**Alternatives to our problem formulation**   Another limitation of our problem formulation is that the *history* of the model pretraining is ignored. Is it sufficient to check whether the evidence data help the *fully* pretrained model? What if a data is truly related to the task but got overfitted during earlier stages of pretraining, so continuing pretraining on it would not change the model? We think this is possible, but only to some extent and with a limited risk, since the huge volume of pretraining data can make it difficult to overfit to all examples related to an arbitrary task in a particular way. Nevertheless, to improve that case, a potentially better solution would be continuing pretraining *each checkpoint* of the model across the whole pretraining procedure. This would share an intuition with some instance attribution methods based on model checkpoints (Pruthi et al., 2020). However, we lack the resources to perform such experiments in this work, so we defer that to future research.[22]

## B    ETHICS STATEMENT

One ethical consideration and a limitation of our approach is the large computational cost, and consequently the environmental impact caused by our computationally-expensive method (Strubell et al., 2019). On our machine with 8 Nvidia A40 GPUs, the full 20 epochs of each ORCA experiment would take about 7 days in total. The long computing time is partly because we need to calculate per-sample gradients for 4M data points in each epoch, and also because there is no efficient way to calculate the per-sample gradients in PyTorch at the time of our implementation.[23] However, our goal is to develop a research prototype (which can be optimized in the future) that will enable opening up the black box of large language models. Insights into their pretraining data will potentially lead to positive impacts—removing problematic data sources, demoting spurious correlations, and alleviating other ethical issues caused by our current inability to interpret decisions of large language models.

## C    CONTINUED DISCUSSION ON EXPERIMENTAL SETUP

**The sample size $t$ in the embedding kNN baseline**   For the embedding kNN baseline, we sample $t = 1000$ task examples from $D^{task}$ and find $k = \{10, 20, 50, 100\}$ most similar pretraining data to each of the task example. This $t$ should already be large enough since even with the smallest $k$, $t \cdot k$ is well over $|S|$ (meaning that we need to downsample anyway). We did not use the entire $D^{task}$ because IMDB has 25K examples, and the calculation has a heavy requirement on the storage and memory. Nevertheless, to check whether the current $t$ is adequate, we experiment with $t = 8000$ and $t = 9800$ for IMDB and MNLI respectively. The performance is 74.82 and 45.04, no better than the $t = 1000$ performance in the main evaluation.

---

[22]With unconstrained resources, one can extend the checkpoint proposal and even measure Shapley values (Shapley, 1951) of the data, an equitable valuation method (Ghorbani & Zou, 2019).

[23]This may be possible with the latest release of functorch (He & Zou, 2021). We plan to work on it and expect a significant speedup.

**Exact optimizer steps based on $S$**   For all of our experiments, we have $|S| \leq 2000$ and a batch size of 16. We additionally held out 5% of $S$ during optimization as a sanity check for the LM loss. Therefore, *at most 119* optimizer steps were applied in all of the experiments.

## D    USING THE TEST SET AS $D^{\text{TASK}}$

The use of test set over training set is our deliberate choice in this work.[24]   There are two main reasons.

- We assume that the task's test data is the only instantiation of the task, since we are interpreting a model deployed in a zero-shot setup. We cannot assume the availability of a "training set" in such scenario.
- More importantly, this work is about interpretability, not absolute model performance. We are interpreting why the model can achieve a good *test* performance, not a good *training* performance. Similar to other interpretability research, no matter feature attribution or instance attribution (Ribeiro et al., 2016; Koh & Liang, 2017), we must use the test data, since we are interpreting the model's behavior on test examples exactly. This may also help us reveal the data artifacts inside the test set (Han et al., 2020).

That being said, we do have sanity checks at the place we use test data in ORCA, that the effect of using the training set gradient would be very close. For all 20 intermediate stages, the cosine similarity between the test set and training set gradients is $0.977_{\pm 0.023}$ for IMDB and $0.947_{\pm 0.042}$ for MNLI, in a range of [-1,1].[25]

## E    CONTINUED DISCUSSION ON MAIN EVALUATION

**Importance of selecting $S$ in several iterations**   While not shown in the main evaluation table, the *first* epoch of ORCA without later iterations ($|S| = 100$) would actually *hurt* the performance (57.39 and 37.43 for IMDB and MNLI respectively). This is due to the imbalance of the selected pretraining data, favoring only one label in the task. ORCA's iterative selection strategy in this case is essential, a difference from the instance attribution methods in previous interpretability research.

**Calibration**   While not a focus of this work, prompt-based language models can be improved with calibration techniques such as $\text{PMI}_{\text{DC}}$ (Holtzman et al., 2021). We did not use such calibration in our work because our pilot study shows a rather even prior distribution among the labels—applying $\text{PMI}_{\text{DC}}$ on our model (LM, template, verbalizers) in IMDB yields a less than 1% performance improvement.

**Hyperparameter search**   Due to the high computing cost mentioned in §B, we did not perform hyperparameter search in our ORCA experiments (whereas we did a search for the embedding kNN baseline). It is possible that some other sets of ORCA hyperparameters can lead to better performance numbers than those in the main evaluation table. We will release all of the code, experiment scripts, and data at `ANONYMIZED` upon publication.

## F    ADDITIONAL ANALYSIS RESULTS

In Figure 3, we show the MAUVE similarity analysis on MNLI, accompanying the MAUVE analysis on IMDB in the main paper.

---

[24]Also note that all of the investigated algorithms, ORCA and the baselines, have the same $D^{\text{task}}$ setup.

[25]These numbers mean the effect of using training and test gradients is extremely similar, noting that the dimension of the gradients is very large (340M). Given two random vectors $a$ and $b$ at this dimension, $Pr[|\cos(a,b)| > 0.9] < 1/e^{2.75 \times 10^8}$ (Arora, 2013).

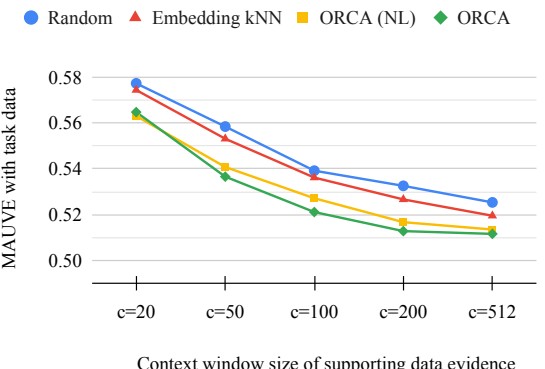

Figure 3: MAUVE similarity on **MNLI**, between the sets of $x_{\text{context}}^{\text{PT}}$ in $S$ and $x^{\text{task}}$.

