# OpenReview forum: "ORCA: Interpreting Prompted Language Models via Locating Supporting Evidence in the Ocean of Pretraining Data"
_ICLR.cc/2023/Conference — Submitted to ICLR 2023_

### Official Review · Reviewer_st4M · 2022-10-24

**Confidence:** 4
**Clarity, Quality, Novelty And Reproducibility:** Please refer to the Strength And Weak…
**Correctness:** 3
**Technical Novelty And Significance:** 2
**Empirical Novelty And Significance:** 2
**Recommendation:** 3

**Strength And Weaknesses:**

Strength:
1. The authors provide a novel metric to locate pretraining data that supports the model’s predictions in a given task.
2. Table 1 shows that using retrieved corpora can effectively improve the effectiveness of LMs in downstream tasks.

Weaknesses:
1. the problem is not well defined. A corpus that improves the effectiveness of a downstream task cannot necessarily be used to explain the predictions of the original language model. While many of the claims in this paper are about the interpretability of pretraining by locating training corpora, the experiments are mostly about performance improvement.
2. If the goal is to improve model effects by some continued pretraining, the related work and experiments in this paper lack comparison with state-of-the-art methods, e.g. dense retriever methods like REALM.
3. The method has some assumptions. However, they are not validated. For example, whether continuing training on the task data directly will improve the original model? Why the gradient of the sample to be found is similar to the gradient of the target task samples?
4. I think the method in this paper cannot be scaled to large LMs. In LMs, there are at least millions of gradients for each sample.
5. Some experiments are confusing. For example, I cannot understand why words like good/bad/yes/no happen to be masked out in Table 3. This is not consistent with the random mask strategy.
6. Table 3 is inconsistent with the intuition of this paper. According to the examples in Table 3, it looks like the model is looking for verbalizer words, instead of the training corpora that explain the model behavior in the downstream task.
7. I suggest that the authors use more downstream tasks for their experiments.

**Summary Of The Paper:**

In this paper, the authors try to locates a very small subset of pretraining data that directly supports the model’s predictions in a given task. The main idea is to use cosine similarity between gradients of the downstream samples and pre-training corpus.  They designed experiments based on 0.5% of the Wikipedia and BookCorpus over IMDB and MNLI.

**Summary Of The Review:**

Overall, I think the problem is not clearly defined. This causes the inconsistencie in the method and experimental descriptions. In addition, there is an issue with the choice of baselines.

---

> ### Author Response · Authors · 2022-11-12
> **Response to Reviewer st4M**
>
> We thank Reviewer st4M for their time and review. However, there seems to be some misunderstanding of our method and we hope to make some clarifications below.
>
> Re: the problem is not well defined. Please note that we have an entire subsection (Section 2.1) in the main paper on the problem formulation (plus Appendix A for further implications and alternatives to the formulation). We discussed how we get to the formulation from conventional instance attribution work like influence functions [1]. Regarding the comment that “the experiments are mostly about performance improvement” -- the purpose of showing the perturbative performance improvement in this work is to verify that the pretraining data subset we located is indeed influential (similar to the leave-one-out verification in influence functions). The reviewer states that “a corpus that improves the effectiveness of a downstream task cannot necessarily be used to explain the predictions of the original language model”. Note that we are not using *any* random training corpora; we use the exact pretraining data the model sees through the pretraining phase, the sole data source the zero-shot LM was exposed to when being deployed to the out-of-distribution downstream task.
>
> It appears that the above misunderstanding of the overall methodology has led to other misperceptions about our work (e.g., the baseline choice). For example, our ultimate goal is not to improve the model performance by an arbitrary type of continued pretraining. The goal of our perturbative finetuning with a small subset of original pretraining data is not to beat the state-of-the-art modeling methods, but for revealing the potential supporting factor for the LM’s success in the zero-shot downstream scenario. More concretely, a retrieval-based REALM indexing into large external knowledge corpus cannot answer why BERT gets non-trivial performance in zero-shot sentiment analysis task.
>
> We also address some separate confusion points below.
>
> Re: whether continuing training on the task data directly will improve the original model. This is conventional finetuning and that’s why we expect an improvement.
>
> Re: why the gradient of the sample to be found is similar to the gradient of the target task samples. This is a part of our algorithm. The intuition behind it is introduced in Section 2.2. Gradient similarity is a classic technique being used in instance attribution methods (e.g., [1], [2]). We also compare with embedding-based methods as a baseline.
>
> Re: why words like good/bad/yes/no happen to be masked out in Table 3 and it is not consistent with the random mask strategy. We still randomly mask words in the pretraining data (for reconstruction), and the important pretraining instances meaningfully have the downstream verbalizer word as the masked word. This effect is more clear in Table 4, and it shows: the reconstruction of verbalizer words and their synonyms in the pretraining data is a part of the reason that zero-shot LMs can perform well on downstream tasks.
>
> Re: scaling to larger LMs. The gradient would indeed be large as the model size grows. However, in those cases it is possible to truncate the gradient (e.g., only taking the gradient over the last M layers of the model). This is not the focus of our current work since our goal is to propose a general methodology to the interpretability problem.
>
> We appreciate the Reviewer’s feedback and time dedicated to reviewing our paper, and we hope that we have clarified some points that we believe were misunderstood. We will be happy  to follow up with further discussion to ensure an accurate evaluation of our problem and contribution.
>
>
> Reference:
>
> [1] Koh and Liang. 2017. Understanding Black-box Predictions via Influence Functions. Proc. ICML.
>
> [2] Pruthi et al. 2020. Estimating Training Data Influence by Tracing Gradient Descent. Proc. NeurIPS.

---

> > ### Comment · Reviewer_st4M · 2022-11-20
> > **response**
> >
> > Thanks to the author for the replies. However, I am not convinced by them. I don't think I misunderstood the authors' paper, but the fact is that the authors' research goals presented in the abstract/introduction section are not consistent with the actual algorithms and experiments performed.
> >
> > 1. the authors' experiment verifies "on what corpus the model can be improved by continuing pre-training". However, the author's research goal in the abstract/introduction section is "from where the model learns task-specific knowledge". Obviously, these are two different questions with different answers. I think the latter is a significantly more important question. Unfortunately, the authors' model and experiments do not really address this question.
> > 2. the problem definition of Sec 2.1 is inadequate for the research objective "from where the model learns task-specific knowledge" introduced by the authors in the abstract/introduction section.
> > 3. The problem of scalability of the method. (a) In LMs, there are at least millions of gradients for each sample. This makes it very expensive to compute similarities, especially for the large corpus used in pre-training. (b) In the experiment, the authors used only 0.5% of the full pretraining data, which is tricky and makes the results unreliable.
> >
> > The authors' other responses (e.g., W5) did not convince me either.
> >
> > Therefore, I would like to keep my rating of this paper.

---

> > > ### Author Response · Authors · 2022-11-21
> > > **Still, likely misunderstanding of the problem formulation**
> > >
> > > It would be very helpful to us if the Reviewer can provide a more **concrete justification/rationale** behind their subjective opinion about our problem formulation. Currently, the Reviewer thinks that what we are doing in the problem formulation is “on what corpus the model can be improved by continuing pre-training” (A), and states this is incompatible with the objective “from where the model learns task-specific knowledge” (B).
> > >
> > > Their statement attempting to describe what we are doing (A) is not actually accurate. More precisely, our research question is: what pretraining data of the model may help the model improve performance on task-specific data. This is connected to B because: (1) We are investigating zero-shot language models where the only training data seen by the model is a large general-purpose pretraining corpus. (2) Without these pretraining data the model is simply randomly initialized thus not task competent. (3) Thus the pretraining data is important to the task competence, but it is very large and we don’t know which aspect of it is important. (4) Therefore, we want to find a small subset of pretraining data that could support the model’s task-specific competence. (5) The level of support is instantiated and verified by continuing pretraining and testing on the task in this work. We also talk about limitations and alternatives to this instantiation in Appendix A.
> > >
> > > We don’t believe that our problem formulation and the objective (B) are “obviously two different questions with different answers”. We welcome further concrete, constructive feedback on this, e.g., which and why any of our rationales (1-5) are wrong.
> > >
> > > As for the scalability of the method, we already responded in our original response under “Re: scaling to larger LMs”. Additionally, using 0.5% of the full pretraining data is indeed a drawback (due to computing), but we believe this is at least an unbiased and a good amount (given the huge total size) of random samples of the full pretraining data, not “tricky and unreliable”.
> > >
> > > The Reviewer further said “the authors' other responses (e.g., W5) did not convince me either”. We would again welcome more concrete critique which we could attempt to address.
> > >
> > > For the points above, we will also appreciate constructive suggestions from the Reviewer on how our approaches can be corrected or improved, and we thank again the Reviewer for their valuable time and feedback.

---

> > > > ### Comment · Reviewer_st4M · 2022-11-22
> > > > **response**
> > > >
> > > > The research objective introduced by the authors in the abstract/introduction section is A: "from where the (pre-trained) model learns task-specific knowledge", while what the authors clarify they are really doing is B: "**continue training** on what pretraining data of the model may help the model improve performance on task-specific data". As I mentioned before, these are two different questions, and problem A makes more sense.It is problem A that responds to the author's discussion in paragraphs 1-3 of the introduction, not problem B.
> > > >
> > > > Problem A may be connected to Probem B. However, I don't think using (1)-(5) to justify the connection is scientific, nor does it advance the understanding of this problem for the research community.

---

> > > > > ### Author Response · Authors · 2022-11-24
> > > > > **Justification needed**
> > > > >
> > > > > Thank you for the follow-up. Could you please clarify **why** do you think our problem formulation (A) “continuing pretraining the pretrained model for one epoch on a small subset of pretraining data (upweighting the subset) and verifying the perturbed model’s task-specific performance” and goal (B) “from where the pretrained model learns task-specific knowledge” are “obviously two different questions with different answers”?
> > > > >
> > > > > For the background, prior work like influence functions rely on similar types of formulations (upweight and validate perturbed performance).

---

### Official Review · Reviewer_uxbN · 2022-10-24

**Confidence:** 3
**Clarity, Quality, Novelty And Reproducibility:** See strengths and weaknesses.
**Correctness:** 3
**Technical Novelty And Significance:** 3
**Empirical Novelty And Significance:** 3
**Recommendation:** 6

**Strength And Weaknesses:**

Strengths:

1. The problem is important, as insights can inform selection of training data for future LLMs. The proposed technique thus has the potential to be useful in many different contexts.

2. The idea is novel. Although gradient-based similarity is widely used, this specific application is new, as far as I know.

3. The paper is very well written: the technique is described clearly and precisely, and the experiments should be easy to reproduce.

Weaknesses:

1. Although the technique is intuitive and easy to understand, it’s not clear to what extent it’s a good approximation of the true quantity of interest, namely performance if the LLM were retrained from scratch on the whole corpus with the identified subset removed. Influence functions obviously get at that more directly, and I think the paper should have included an influence-function baseline. This needn’t be an iterative procedure (which might be difficult to formulate), just the set of individual examples with the greatest influence.

2. The experiments are somewhat limited: only on BERT, and only on two tasks. They are also limited in exploring only tiny subsets. It’s interesting to know that fine-tuning on such a set can provide a good performance boost (at least in the case IMDB), but it would be better to have a sense of where this flattens out as the subset is expanded.


**Summary Of The Paper:**

This paper proposes a method for finding the subset of a large language model (LLM) training corpus that is most responsible for the 0-shot performance of the LLM on a given task. The method involves finding examples whose gradient is similar to the gradient of task-specific supervised data. This is done in an iterative process in which the original LLM is fine-tuned on the examples found during the previous step, in order to account for interactions among examples. Performance is evaluated by fine-tuning the LLM on the identified subset, then measuring task performance.

On IMDB and MNLI tasks with BERT, the proposed method is found to outperform random-sampling and kNN baselines. The most striking finding in the resulting analysis is that most of the valuable examples come from the Books portion of BERT’s training corpus.


**Summary Of The Review:**

The paper is a bit thin on substance, both in terms of theoretical justification and breadth of experimental results. However, it attacks a very central problem, is clearly written and proposes a simple, potentially very useful technique, so I am leaning in favour of acceptance.

---

> ### Author Response · Authors · 2022-11-12
> **Response to Reviewer uxbN**
>
> We thank Reviewer uxbN for their thoughtful review. We appreciate the reviewer for identifying the problem we address as important and our proposed method as novel and potentially useful.
>
> The suggestion of comparing with influence functions makes a lot of sense theoretically, but in practice is prohibitively expensive (infeasible even if we have orders of magnitude more computing resources). The main obstacle is estimating the inverse Hessian term using the LiSSA method (where we have the implementation in pilot study). If we ignore the inverse Hessian term, influence functions become a dot product between model gradients. This would be remotely similar to an one-epoch ORCA, and we show in Appendix E that it hurts the performance (we mention a similar finding in Footnote 2). Apart from influence functions (discussed in Section 2.1 and Footnote 5), we also discuss an alternative to our current perturbative finetuning formulation in Appendix A.
>
> The suggestion on investigating when the performance gain would flatten out as the size of the subset grows can indeed be an interesting analysis for future work. We can already see (roughly) from our Table 1’s trajectory that the performance is plateauing as |S| grows. As for the limited number of tasks and models (BERT on two canonical text classification tasks), we agree this can be improved in future work if one has more resources (please refer to Appendix B).  On our machine with 8 Nvidia A40 GPUs, each single ORCA experiment took 7 days to complete (in the appendix we also explained the reason and potential improvements in the future). We have two ORCA variants and investigated two canonical text classification tasks, which in total already resulted in about 1 month of computing time (not including pilot study verifying reasonable hyperparameters). We work in an academic lab with less access to dense computing resources than in the industry. Our goal is to introduce a novel working methodology addressing a difficult and important research problem of large LM interpretability, and we would be happy if researchers with more compute resources are interested in extending the work in the future.

---

### Official Review · Reviewer_sFZJ · 2022-10-26

**Confidence:** 3
**Clarity, Quality, Novelty And Reproducibility:** The Clarity and Quality is Good!
**Correctness:** 3
**Technical Novelty And Significance:** 3
**Empirical Novelty And Significance:** 3
**Recommendation:** 6

**Strength And Weaknesses:**

Strengths:
This paper proposed a gradient-based method to identify the evidence of the model’s task-specific competence in prompt-based learning based on the general large pre-trained language model. It is interesting.


Weaknesses:
Novelty of the proposed method is somewhat limited.

**Summary Of The Paper:**

The authors want to explore a very interesting question why the prompting learning of large pretrained language models leads to strong performance in a variety of downstream tasks, especially in zero-shot setups? This paper proposed a novel method to identify the evidence of the model’s task-specific competence in prompt-based learning based on the general large pre-trained language model. It uses the gradient information related to the downstream task, IORCA can locate a small subset of pretraining data that is similar to the downstream tasks. This work is an very interesting exploration but not very novelty.

**Summary Of The Review:**

Same with Summary Part.

---

> ### Author Response · Authors · 2022-11-12
> **Response to Reviewer sFZJ**
>
> We thank Reviewer sFZJ for finding our method a “very interesting exploration”. We believe our work addresses an important topic on interpreting why zero-shot prompt-based LMs are effective in downstream tasks, by relying only on the general-purpose pretraining data.

---

### Official Review · Reviewer_NC89 · 2022-10-26

**Confidence:** 4
**Correctness:** 4
**Technical Novelty And Significance:** 3
**Empirical Novelty And Significance:** 3
**Recommendation:** 5

**Clarity, Quality, Novelty And Reproducibility:**

I think that the paper has enough novelty. The writing is ok although not particularly brilliant, and the mathematical formalization is understandable but could be made more clear. The paper does a good job in terms of reproducibility: I did not miss any critical experimental detail, and the authors promise that the code and data will be released.

**Strength And Weaknesses:**

STRENGTHS
- The paper tackles an important problem (understanding how pretrained models are able to perform downstream tasks in a zero-shot fashion), and the approach followed is original and technically sound.
- The finding that the context of the supporting data evidence is not particularly similar to the task input data is interesting and not obvious.

WEAKNESSES
- While I really like the general idea of the paper, the analysis itself is rather shallow. The main findings are (i) BookCorpus is more relevant than Wikipedia despite the latter being larger, (ii) pretraining examples masking the downstream verbalizers are the most relevant, and (iii) the most relevant pretraining examples are not particularly similar to the downstream task examples. While (iii) is interesting, (ii) is not surprising at all, and more analysis is necessary to interpret (i). For instance, it could be that the verbalizers (e.g. good and bad) occur more frequently in BookCorpus, in which case (i) would be implied by (ii) and not surprising at all.
- I am not satisfied by the evaluation of the proposed approach in Section 4. ORCA outperforms all baselines in Table 1, but that is not surprising at all, as it is the only approach using labeled downstream data, and it works by picking examples from the pretraining corpus whose gradient is similar, therefore approximating gradient descent in downstream data. Some basic baselines are missing (e.g. only picking examples that mask the downstream verbalizers), and it would be useful to have a variant finetuning on the downstream task as an upper-bound to put the numbers in perspective. It would also be useful to finetune another pretrained model (e.g., RoBERTa) in the same set of examples (selected based on BERT): if these examples are truly supporting the task in question, I would expect them to be helpful for all models. In addition, the results in Table 2 are negative: even if the authors claim that they get an improvement on IMDB, this is likely not significant, as this applies to only one variant of the proposed method and even in that case the improvement is only 0.27 points while the standard deviation is 0.65. This is puzzling and deserves some more discussion: does this imply that all pretraining data is equally useful when using prompt-tuning?
- The manual analysis of the examples is dissapointing. We are only shown a single example for each task and verbalizer. As the authors say, it is not clear at all how the examples mined for MNLI can be useful to learn entailment. The authors say that the examples mined for IMDB do express sentiment, but I think that even that is questionnable (the second one does, but the first one is talking about the genetics of bacteria). I would have liked to see more examples in the appendix and a more systematic analysis of them, with the aim to understand how interpretable the examples are.
- The evaluation is limited to two tasks and a single model, using a very small fraction of the pretraining corpus. While I understand that the authors were constrained by compute, this cannot be use as a cheap excuse for everything. I would have liked to see more extensive experiments and, if truly not feasible due to compute constraints, the authors should at least report the compute required for each experiment to justify this (e.g. if each experiment takes a month on their hardware I would understand not having more, but if it takes a few hours this wouldn’t be a valid excuse).

**Summary Of The Paper:**

This paper proposes a new approach to find a subset of the pretraining corpus that supports BERT’s zero-shot predictions in a given task. This is done by iteratively finding pretraining examples whose gradient is the most similar to that of the downstream task examples. The technique is used to analyze the performance of BERT on IMDB and MNLI, finding that (i) BookCorpus is more relevant than Wikipedia despite the latter being larger, (ii) pretraining examples masking the downstream verbalizers are the most relevant, and (iii) the most relevant pretraining examples are not particularly similar to the downstream task examples on the surface, showing that the model is not relying on pure memorization.

**Summary Of The Review:**

I think that this is a borderline paper, perhaps leaning a bit more on the negative said. It tackles an important problem from an original angle and is technically sound, but the analysis presented is rather shallow.

---

> ### Author Response · Authors · 2022-11-12
> **Response to Reviewer NC89 [1/2]**
>
> We thank Reviewer NC89 for their detailed and helpful review. We are glad that the Reviewer thought that our method is original and technically sound, the research problem is important, and the overall paper is novel. Below we make clarifications that we believe are key to some of the raised concerns about the evaluation and analysis sections. We split our response to two parts due to the default character limit.
>
> Regarding the evaluation:
>
> Re: fair comparison with baselines. The Reviewer mentioned that ORCA outperforming all baselines in Table 1 is not surprising at all, since it’s the only approach using the labeled downstream (task) data. However, we would like to clarify that only the random baseline is not using the labeled task data, and all other baselines are using the labeled task data (more specifically, they use both the context and the label/ground-truth verbalizer of the task data). One essential baseline, Embedding kNN (introduced in Section 3.1) calculates embeddings with the ground-truth verbalizer passed in (more details in Eq. 6, in short we supply the ground-truth verbalizer to the task data and take the hidden representation at that position; for the pretraining data we extract representation individually at every possible position). This is because we want to know which subset of pretraining data is most similar to the *labeled* task data, rather than an unlabeled task context. This makes our baseline different from conventional kNN methods in, say, [1] (they cannot use labels since they are doing inference, and we are doing interpretation). In fact, in our pilot study, we tried using the unlabeled embedding representation as well and it led to a large degradation in performance so we omit these experiments from the paper.
>
> Two additional suggestions were given regarding the evaluation: (1) to include a method picking examples that mask the downstream (task) verbalizer, and (2) to test the supporting data evidence (SDE) found for BERT on other models (and expect them to be helpful for all other models). However, these overlook one property of the model “interpretation” that we are seeking: the supporting data evidence should not only condition on the downstream task, but also condition on the specific model (parameters) we are inspecting (since we are interpreting this specific model). That being said, we also think the matching in downstream verbalizer words is an important factor. That’s why we have the embedding kNN baseline calculated in a way that extracts the verbalizer’s representation. In fact, in Table 4 (especially the IMDB result) we see a majority (94.8%) of data discovered by the Embedding kNN baseline has the exact downstream verbalizer word as the masked token. Still, our proposed method ORCA outperforms the baseline.
>
> Re: negative results in Table 2. In Table 2, our secondary results on interpreting LMs with prompt tuning, indeed the improvement on IMDB may not be statistically significant given the deviation. We will make it clear in the next version that this part of the paper reports a negative result. We intentionally chose to report this result since we believe that although it’s negative it is still interesting. As we explained in the last paragraph of Section 4, it is possible that the tuned prompt (could be viewed as a form of finetuning already) makes the LM highly specialized towards the task, using the in-task training data. Therefore, the *additional* signals in the pretraining data that are useful to the task can be scarce. We do not make claims about which or whether all pretraining data is useful in the case of prompt-tuning. It can be a good future work direction, but our main goal in this paper is still to interpret why the zero-shot LM is an effective model for some tasks.
>
> Re: the scale of the experiments (number of tasks and models). We already included in Appendix B of our paper the exact computing resource used for the current experiments. On our machine with 8 Nvidia A40 GPUs, each single ORCA experiment took 7 days to complete (in the appendix we also explained the reason and potential improvements in the future). We have two ORCA variants and investigated two canonical text classification tasks, which in total already resulted in about 1 month of computing time (not including pilot study verifying reasonable hyperparameters). We work in an academic lab with less access to dense computing resources than in the industry. Our goal is to introduce a novel working methodology addressing a difficult and important research problem of the prompted large LM’s interpretability, and we would be happy if researchers with more compute resources are interested in extending the work in the future.

---

> ### Author Response · Authors · 2022-11-12
> **Response to Reviewer NC89 [2/2]**
>
> (Please note that this is a second part of our response)
>
> Regarding the analyses:
>
> Re: surpriseness of analysis results. The reviewer raised a concern that among our three analyses of the supporting data evidence, the second one regarding the masked token is “not surprising at all” and subsequently the first one about the importance of sub-corpus is also “not surprising at all”. We respectfully disagree (for example, Reviewer uxbN instead finds the result “striking”). We emphasize that our results and analyses are novel; even if some readers will find these findings intuitive, no prior work was able to interpret predictions of prompted large LMs similarly backed up with experiments.
>
> As for the first IMDB qualitative example, indeed it is about bacteria/biology, but expressing sentiment over that sub-span is not mutually exclusive with the whole paragraph’s topic (separately we showed in the third analysis that the topic/context of the SDE is usually distant to that of the task). Another qualitative observation from the examples is that they contain sentimental words from both positive and negative sides: awfully, lucky, nice, sloppy, good (in the first IMDB example), better, bad (in the second IMDB example). In IMDB data, the movie reviews often contain words from both sides as well. Showing these quantitatively is not trivial and can be an interesting future work, but the focus of this work is to introduce ORCA as a method to find SDE and showcase what type of information we can derive from it.
>
> We include only 5 examples in Table 3 due to space constraints. We will show in the appendix in the next version more SDE examples along with the task examples. We did not cherry pick the examples that we presented in Table 5; the only filtering we have is that the masked word of the displayed examples is within the verbalizer word set (see Table 4 for the percentage of satisfying examples), for a more straightforward understanding for the audience.
>
> We appreciate that the reviewer found the third analysis interesting (supporting data evidence from pretraining data is not similar to downstream task examples on the surface, indicating the model may not be relying on pure memorization).
>
>
> Reference:
>
> [1] Khandelwal et al. 2020. Generalization through Memorization: Nearest Neighbor Language Models. Proc. ICLR.

---

### Decision · Program_Chairs · 2023-01-20

**Decision:**

Reject

**Justification For Why Not Higher Score:**

While the problem is interesting, it is done in a superficial manner with concerns about the task formulation and experimentation.

**Justification For Why Not Lower Score:**

N/A

**Metareview: Summary, Strengths And Weaknesses:**

This paper attempts to interpret the performance of prompt-based model. The authors proposed to use the gradient of a downstream task to identify the samples that contribute the most to the task. Fine-tuning with theses samples will further improve the task performance, although the samples may not resemble the task.

All reviewers agree that the problem is interesting. However, concerns are raised about the formulation and experiments. We agreed that the paper may not meet the bar of ICLR.

**Summary Of Ac-Reviewer Meeting:**

We did a virtual meeting on Nov 28. Reviewers NC89, st4M, and myself participated in the meeting. Reviewer uxbN accepted the meeting invited but was occupied by his work. Reviewer sFZJ could not make it (as no single time works for all of us).

We believe the problem is interesting. However, it's done in a superficial manner with concerns about the task formulation and experimentation. We agreed that the paper may not meet the bar of ICLR.